# Research on Network Patterns and Influencing Factors of Population Flow and Migration in the Yangtze River Delta Urban Agglomeration, China

**Xuewei Wang** [1,2,*] **, Shuangli Ding** [1,*]**, Weidong Cao** [1]**, Dalong Fan** [1] **and Bin Tang** [3]

[1]   School of Geography and Tourism, Anhui Normal University, Wuhu 241002, China;
    weidongwh@163.com (W.C.); fandl001@ahnu.edu.cn (D.F.)
[2]   School of Geography and Planning, Cardiff University, Cardiff CF 10 3WA, UK
[3]   School of Architecture, Southeast University, Nanjing 210096, China; tangbinzsj@hotmail.com
[*]   Correspondence: wangxw697@ahnu.edu.cn (X.W.); aliceding375@hotmail.com (S.D.)

**Abstract:** Through the construction of a population flow and migration relationship matrix, this paper analyzes population flow and migration in the Yangtze River Delta urban agglomeration during the Spring Festival travel rush and daily period. This paper also studies the urban network spatial structure characteristics and the influencing factors from the perspective of inter-provincial population flow and migration. The results show the following: (1) as a central city, Shanghai has a significant siphon effect, with Suzhou, Nanjing, Hangzhou, Ningbo, Wuxi and Changzhou accumulating 86.95% of the incoming population. The Shanghai–Jiangsu cross-border floating population is active and accounts for 40.83% of the total mobility scale in the same period. The population flow and migration network in the Yangtze River Delta urban agglomeration shows obvious hierarchical characteristics. The secondary network relationship during the Spring Festival travel rush is the main migration path, while the first-level network relationship in the daily period is the main flow path. (2) Three indicators, namely, the network density, mean centrality, and control force based on the population flow and migration, consistently show that the Yangtze River Delta urban agglomeration network presents a strong connection state with the formation of a local cluster structure, highlighting that the city tightness in terms of population flow and migration also has dual attributes, which refers to "the restriction of the geographic space effect" and "overcoming the friction of space". (3) Economic scale, political resources, industrial structure, and the historical basis are important factors influencing the formation of population flows and migration networks. Employment opportunities and labor wages are key guiding factors of the population migration direction, and spatial distance is a conditional factor influencing the formation of population flows and migration networks. The inter-provincial boundary, temporal distance, and transboundary frequency are the decisive factors for the formation of network patterns of population flow and migration.

**Keywords:** population flow; population migration; network structure; influencing factors; inter-provincial boundary; quadratic assignment procedure; Yangtze River Delta urban agglomeration

---

## 1. Introduction

An urban network is a kind of urban relationship generated in a relatively unconstrained space–time scale by flow factors such as production, service, information and traffic, etc. [1]. A multi-scale urban network and the consequent difference research methods are the advantages of an urban network in the study of urban systems [2]. Urban network studies based on the perspective of urban agglomeration have attracted a great deal of attention. At present, scholars

have revealed the spatial structure characteristics of urban agglomerations from the perspectives of structure compactness [3], stability [4], economic connection [5], factor analysis [6], the man–land relationship [7], and the multi-center network development mode [8]. Studies on the Yangtze River Delta, the urban agglomeration with the highest development maturity in China, paid more attention to regional cooperation, regional spatial coordination, and development path exploration [9,10]. As the integration of Yangtze River Delta has become a national strategy, theoretical and empirical studies on urban innovation [11], isomorphism [12], the metropolitan area [13], and spatial connection have increased [14,15]. However, integrative development is faced with the dual problems of constructing a regional network and eliminating the restriction of boundaries. On the one hand, creating networks is necessary for the integrated development of urban agglomerations. Under the background of flow space, various high-speed infrastructure and information networks make the connections between cities extend beyond the cognitive limits of the previous central place era and form a network relationship of sharing space and time between cities of multiple spatial scales and levels. Due to the regional division of labor and functional supplements, spatial flow, as the essential content of urban spatial connection, becomes a spatial vector with direction and intensity, and its change will lead to a change in the regional connection network, thus leading to the generation of a spatial difference [16]. As the network relationship between cities has been gradually strengthened, the urban flow gradually blurs administrative boundaries, which makes the relative position, relative correlation, and interaction of cities within the region present new characteristics [17]. On the other hand, the inter-provincial boundary is a realistic problem facing the integrated development of urban agglomerations. Since the beginning of the 21st century, boundary research has gradually emerged, mainly focusing on boundary trade [18,19], market segmentation [20], factor costs [21,22], capital differences [23,24], cross-border cooperation [25,26], structure estimation [27], and regional development and coordination [28–33], etc. However, because boundary discussions are mostly concentrated in the economic field, and the relative flow data between administrative regions are difficult to obtain, there are few studies based on the perspective of network relationships of factor mobility. With the continuous advancement of globalization and regional integration, the regional network, as a key part of the country and global network, makes the development of internal migration networks become the important component of the transnational network, as well as the basic unit of a transnational network's construction. So far, the perspectives of boundary research have constantly been diversified as follows [34]: economic geography regards economic activities as homogeneous elements and discusses the differentiation of the phenomenon of cross-regional economic development caused by administrative boundaries [35]. Urban geography regards urban attributes as homogeneous elements and discusses the differentiation phenomenon of cross-regional structure development caused by administrative boundaries [36–38]. Political geography regards political entities as homogeneous elements and discusses the differentiation of trans-regional cooperation mechanism and policy governance caused by administrative boundaries [39,40]. The problem is that the aforementioned research only focused on the phenomenon of regional differentiation caused by the objective existence of boundaries, including a lack of exploration in the construction paths and development mechanisms of the international network embedded in national boundaries with the background of globalization, as well as regional networks embedded in inter-provincial boundaries with the background of regional integration. As a unique product of administrative divisions in China, the objective existence of inter-provincial boundaries makes the regional differential size become a key factor influencing the formation and development of urban network relationships under the boundary theory system.

Population migration has been one of the largest and most far-reaching geographical processes since China's reform and opening up [41]. Social and economic development, reform of the household registration system, and construction of high-speed transportation facilities have promoted population flow and migration. On the one hand, the phenomenon of population flow in the daily stage, i.e., population flow as a process of the spatial reallocation of production factors, where its occurrence and

development is an important driving force to promote social and economic development [42], is usually accompanied by social and economic prices. In an urban network, the vertical, horizontal, formal, and informal links between various enterprises and organizations oriented by value creation constitute the basis for the existence of the population mobility network. On the other hand, the phenomenon of population migration during the Spring Festival travel rush is known as the most spectacular periodic population migration in human history. It has a completely different scale and characteristics as compared to daily population movement. In 2015, about 2.809 billion people migrated during the Spring Festival travel rush in China, among which the most significant part was the large-scale migration of the labor force from rural areas to cities (research on the characteristics and mechanisms of population migration based on these data can provide a good reference for similar research on characteristics of rural–urban migration in the process of industrialization in India, Mexico, Brazil, and other countries), which is also known as the "rite of passage" for rural young people [43]. In general, the regional economic development is not balanced, and the spatial disharmony between population distribution and economic activities leads to the differentiation of population and employment space. Cities in economically developed regions are more attractive than those in economically backward regions, so people tend to choose cities with more developed economies, more job opportunities and better infrastructure. Under the influence of traditional festival culture and customs, a large floating population flows from work cities to their home cities before the Spring Festival, forming the "return migration". After the Spring Festival, people return from their home cities to their work cities, forming the "rework migration". As a major social phenomenon unique to China, Spring Festival travel has become an important perspective to study the migration of floating populations [44–46].

Population mobility is a large-scale group behavior in space and time, and the estimated number of the floating population in China is very sensitive to the counting method used. Studies based on census data show that traditional destination surveys fail to account for more than a third of the floating population, where mainly young people and rural registered resident holders are included [47]. The arrival of the era of big data makes it possible to obtain human movement patterns based on massive spatiotemporal trajectories of individual granularity. The development of global positioning systems (GPS), location-based services (LBSs), and other technologies has provided technical support for the observation of spatial and temporal characteristics of large-scale population behaviors. These methods have been widely used in the study of human space trajectories and space interactions based on stream data [48]. The large number of mobile computing devices with positioning functions and the large amount of spatiotemporal labelled data with individual granularity (including mobile phone call data, bus card swiping records, social network check-in data, taxi tracking, bank card swiping records, etc.,) makes it possible to track individual movements for a long time with high precision and efficiency [49]. The study of individual or group behaviors based on big data, spatial cognitive rules, spatial behaviors and interaction patterns contained in activities is used to support cross-border flow decisions regarding social and economic factors, and this has become the frontier of geographic science research.

To sum up, this paper studies urban network structure characteristics and their influencing factors in the perspective of Spring Festival travel and daily population flow and migration. The research goals of this paper are the following: firstly, to carry out urban network construction and structural characteristics analysis based on the perspective of population flow and migration; secondly, to explore the influencing factors of the formation and development of urban networks based on a quadratic assignment procedure (QAP), improving and testing the influencing factor groups of the formation and development of the network structure. Existing research of network structure influence factors mainly include the economic scale level, political resources, facilities, service quality, living environment, and distance, etc. However, network construction is a process of breaking through and covering administrative boundaries. Therefore, how boundary variables affect the formation and development of a network is a topic worthy of discussion. Based on that, this paper adds three factors, namely time distance, and the administrative boundary, as necessary supplements to the existing element group because of China's special administrative divisions and transboundary frequency. At the same time,

constructing a quantitative analysis system between the network matrix and the variable matrix, the rationality and necessity of adding new elements is tested through the matrix correlation and regression analysis, as well as proving that the relationship between the urban network and inter-provincial boundaries is not a "false relationship". On the one hand, the research perfects the theoretical research system of an urban network and provides a scientific basis for the exploration of regional spatial networkization paths. On the other hand, it can provide a typical case analysis for research on the formation and development mechanisms of cross-boundary networks based on element flow under the background of globalization and regional integration.

## 2. Data and Methods

### 2.1. Research Area

The Yangtze River Delta urban agglomeration is one of the most developed regions in China's social economy and is also an important intersection of the Belt and Road Initiative and the Yangtze River Economic Belt. The economic hinterland of the Yangtze River Delta urban agglomeration is vast, with a modern river and seaport and airport groups. The expressway network is sound, the density of trunk lines of public and railway traffic is the highest in the country, and a three-dimensional and comprehensive traffic network has basically taken shape. In 2016, according to the "Yangtze River Delta Urban Agglomeration Development Plan", the Hefei metropolitan area in Anhui province was included into the Yangtze River Delta urban agglomeration, expanding its scope from "two provinces and one city" to the current "three provinces and one city". In terms of the geographical space, 26 cities, including Shanghai, Nanjing, Hangzhou, and Hefei (Figure 1), cover an area of 217,700 square kilometers. In 2019, the GDP of the Yangtze River Delta region was 19.74 trillion yuan, with a total population of 156 million, which accounts for 2.2 percent of the country's total land area, producing 19.92 percent of the country's production value and representing 11.11 percent of the population.

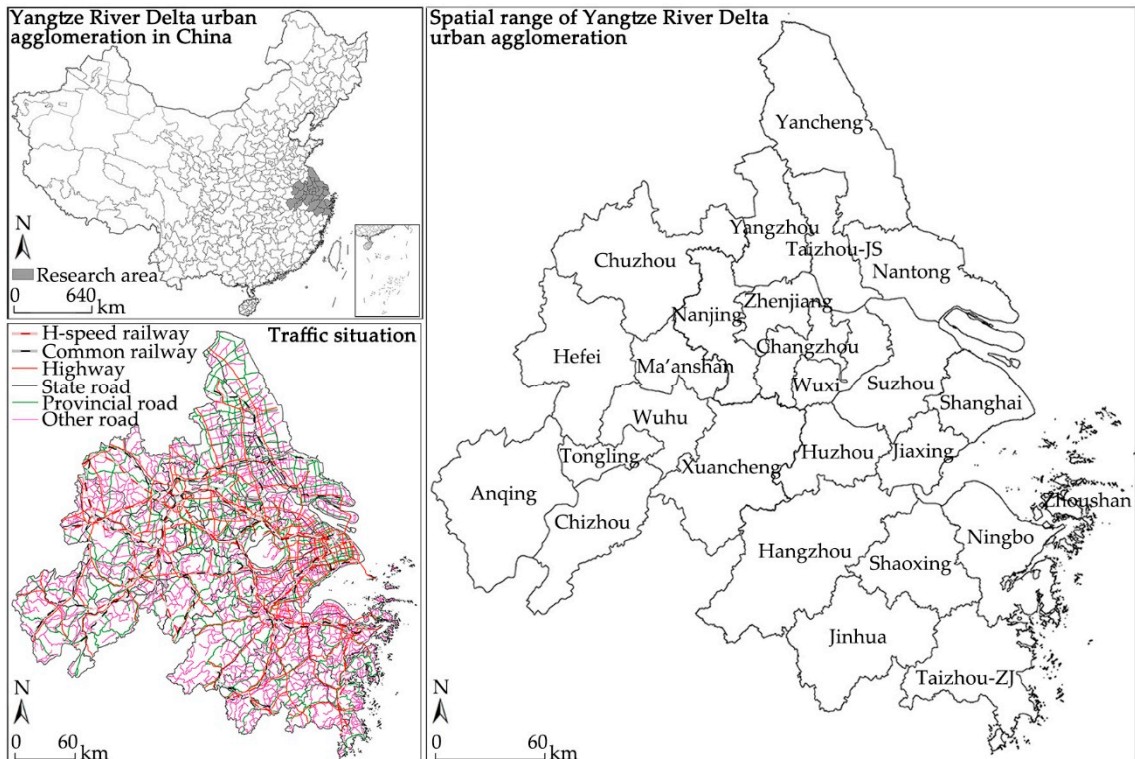

**Figure 1.** Research area scope map.

## 2.2. Data Sources

The migration data of the population involved in the study were sourced from the migration data of Baidu Maps. Based on LBS technology, Baidu migration maps the trajectory of population flow via the location information of mobile phone users and shapes the starting and ending nodes and relationship strengths generated in the process of population flow. The data of population migration consists of two parts: one is the Spring Festival transport data, which represents the characteristics of the net population migration in the Yangtze River Delta urban agglomeration; the second is the daily data, which is used to characterize the spatial network characteristics of population mobility in the Yangtze River Delta urban agglomeration. The study time was divided here into 40 days of Spring Festival travel from 7 February to 18 March in 2015 (the officially defined period of Spring Festival travel in China is generally 40 days, namely, 15 days before and 25 days after the Spring Festival) and 43 days of daily travel from 19 March to 30 April in 2015, which is not a major holiday period. During the 2015 Spring Festival travel rush, the Baidu migration platform identified 81.74 million instances of inter-provincial migration, involving 22,420 directional population migration paths. After processing, the scale of population migration was found to be 466,400. In the daily 43 days, Baidu migration identified 38.54 million people moving between provinces. Through the Spring Festival travel and daily Baidu migration data, the population flow and migration matrices among 26 cities in the Yangtze River Delta urban agglomeration were constructed, which serves as the basis for the analysis of the network structure characteristics of the Yangtze River Delta urban agglomeration. In addition, the index data related to the social and economic development of cities involved in the study were derived from the statistical yearbooks of the provinces (or cities) and statistical bulletins of the national economic and social development of cities.

## 2.3. Research Methods

### 2.3.1. Social Network Analysis

According to the analysis method of a social network, a society is a huge network composed of various relationships and each actor is a node in the network. By studying the network relationships, the relationship between individuals can be grasped so as to reveal the integration and hierarchies of the network. This paper uses the social network analysis method to quantitatively analyze the network structure characteristics of population migration in the Yangtze River Delta urban agglomeration from three aspects, namely, the network density, centrality, and control force.

Network density refers to the degree of the closeness of connections between nodes in a relational network, which is obtained by dividing the number of actual connections in the network by the number of theoretical connections. The calculation formula is the following:

$$D = \sum_{i=1}^{k} \sum_{j=1}^{k} d(n_i, n_j) / k(k-1) \tag{1}$$

where $D$ is the network density, $k$ is the number of nodes in the city, $n_i$ and $n_j$ are any two node cities, and $d$ is the strength of the relationship between the two cities, which is represented by the population flow weight in this paper.

Centricity reflects the population of a region or urban agglomeration in terms of the resource capacity and it can reflect a city's ability to absorb foreign population inflows, with net ingoing values used to measure population during the Spring Festival. Based on the Spring Festival time schedule and overall observation of the data, this determined that the segmentation point of return and rework population migration is midnight on the 24 February. Based on the time arrangement and overall observation of the data, the starting and ending times for return migration were determined as 7–23 February before the festival, with the dominant direction of migration by the Yangtze River Delta urban agglomeration being internal to the external diffusion (diffusion, D). The rework migration after

the festival occurs from 24 February to 18 March, and the leading direction of population migration is aggregation from the outside to the inside of the Yangtze River Delta urban agglomeration (aggregation, A). The calculation formula of this is given as follows:

$$Centrality_i = \sum NI_{i\_dayA} - \sum NI_{i\_dayD} \tag{2}$$

In the formula, $Centrality_i$ represents centrality of a city. When $Centrality_i > 0$, this indicates that city $i$ is the move-in place of population migration during the Spring Festival travel rush, while when $Centrality_i < 0$, city $i$ is the move out place of population migration during the Spring Festival travel rush. $NI_i$ is the flow scale of the urban population in city $i$ and $NI_{i\_dayA}$ is the net flow scale of urban population in city $i$ before the festival, while $NI_{i\_dayD}$ is the net flow scale of urban population in city $i$ after the festival.

The control force is the total intensity of the inter-provincial population flow during daily life, which can reflect the active degree of the population flow in a city. It can be measured by the total flow scale. The calculation formula of this is given as follows:

$$Power_i = R_i^T + R_i \tag{3}$$

where $R_i$ is the inter-provincial flow matrix of population and $R^T{}_i$ is the $R_i$ -transpose matrix. According to the calculation demand, this paper constructs the matrix of population resettlement intensity of 26 cities in the Yangtze River Delta urban agglomeration, the whole country, and the urban agglomeration itself.

### 2.3.2. Quadratic Assignment Procedure (QAP)

The QAP can be used to study the relationship between relationships, namely, to study the correlation and regression between matrices. The QAP needs to construct the city attribute relationship matrix. If a certain attribute of city $i$ and city $j$ is the same, the encoding value at the intersection of $i$ and $j$ in the attribute relation matrix is 1, otherwise the encoding value is 0. Based on a directed, weighted asymmetric matrix during the Spring Festival, the daily time undirected, weighted symmetric matrix, and the city attribute matrix, matrix correlation analysis (QAP correlation) and matrix regression analysis (QAP regression) were performed. The former is used to test the correlation relationship between a population flow network matrix and city property matrix, such as the border area or transboundary frequency, and the latter is used to quantitatively identify the influence of the city property matrix on the population migration network matrix.

## 3. Analysis of Urban Network Pattern

### 3.1. External Connection of Population Resettlement

During the Spring Festival travel rush, the total net immigration population was 11.33 million people, accounting for 24.3% of the national total. Twelve cities, including Shanghai, Suzhou, and Hangzhou, saw a net migration of more than 10 million people, while 13 cities, including Zhoushan, Tongling, and Hefei, saw a larger emigration than the number of people moving in (Table 1). In addition, the spatial characteristics of population migration between the Yangtze River Delta urban agglomeration and the national cities during the Spring Festival travel rush show significant spatial heterogeneity and hierarchical characteristics (Figure 2). Cities in the Yangtze River Delta region and cities in the near region have a close relationship with population migration. The values of the network relationships at the three levels were 17, 363, and 4376, and the scales of the carrying population migration were 2.31, 9.08, and 3.9 million people, respectively. In daily periods, population flow is associated with 35.2 cities on average in the Yangtze River Delta urban agglomeration, and each associated path carries 62,400 people. The scale of daily population flow in Shanghai is 2.16 times that in Suzhou and four times that in Hangzhou and Ningbo, with a prominent siphon effect. In terms of the spatial structure, this also

presents obvious hierarchical characteristics (Figure 3). The numbers of the network relationship at the three levels are 14, 163, and 5746, and the scales of the carrying population flow are 4.75, 3.95, and 3.00 million, respectively. The comparison shows that the number of related cities in the Yangtze River Delta urban agglomeration in the daily period is greater than that during the Spring Festival travel rush, and the radiation scope is wider.

**Table 1.** Net migration scale of cities in the Yangtze River Delta urban agglomeration (ten thousand people).

| A Sequence | City | Net Migration Scale | Flow Scale | A Sequence | City | Net Migration Scale | Flow Scale |
|---|---|---|---|---|---|---|---|
| 1 | Shanghai | 479.23 | 509.31 | 14 | Zhoushan | 0.12 | 28.15 |
| 2 | Suzhou | 220.85 | 235.47 | 15 | Tongling | 1.94 | 22.12 |
| 3 | Hangzhou | 157.93 | 129.32 | 16 | Hefei | 5.13 | 19.46 |
| 4 | Ningbo | 118.37 | 123.55 | 17 | Ma'anshan | 5.52 | 18.41 |
| 5 | Jinhua | 86.05 | 66.57 | 18 | Yangzhou | 6.96 | 18.36 |
| 6 | Nanjing | 72.51 | 56.35 | 19 | Wuhu | 10.58 | 16.49 |
| 7 | Wuxi | 64.87 | 48.17 | 20 | Chizhou | 13.26 | 15.62 |
| 8 | Jiaxing | 47.80 | 45.15 | 21 | Nantong | 13.95 | 15.59 |
| 9 | Taizhou-ZJ | 38.15 | 44.91 | 22 | Taizhou-JS | 14.78 | 14.11 |
| 10 | Changzhou | 29.34 | 41.21 | 23 | Xuancheng | 23.36 | 11.12 |
| 11 | Shaoxing | 28.70 | 37.74 | 24 | Yancheng | 26.46 | 10.65 |
| 12 | Huzhou | 13.90 | 36.23 | 25 | Chuzhou | 45.93 | 4.86 |
| 13 | Zhenjiang | 4.04 | 30.82 | 26 | Anqing | 60.74 | 1.42 |

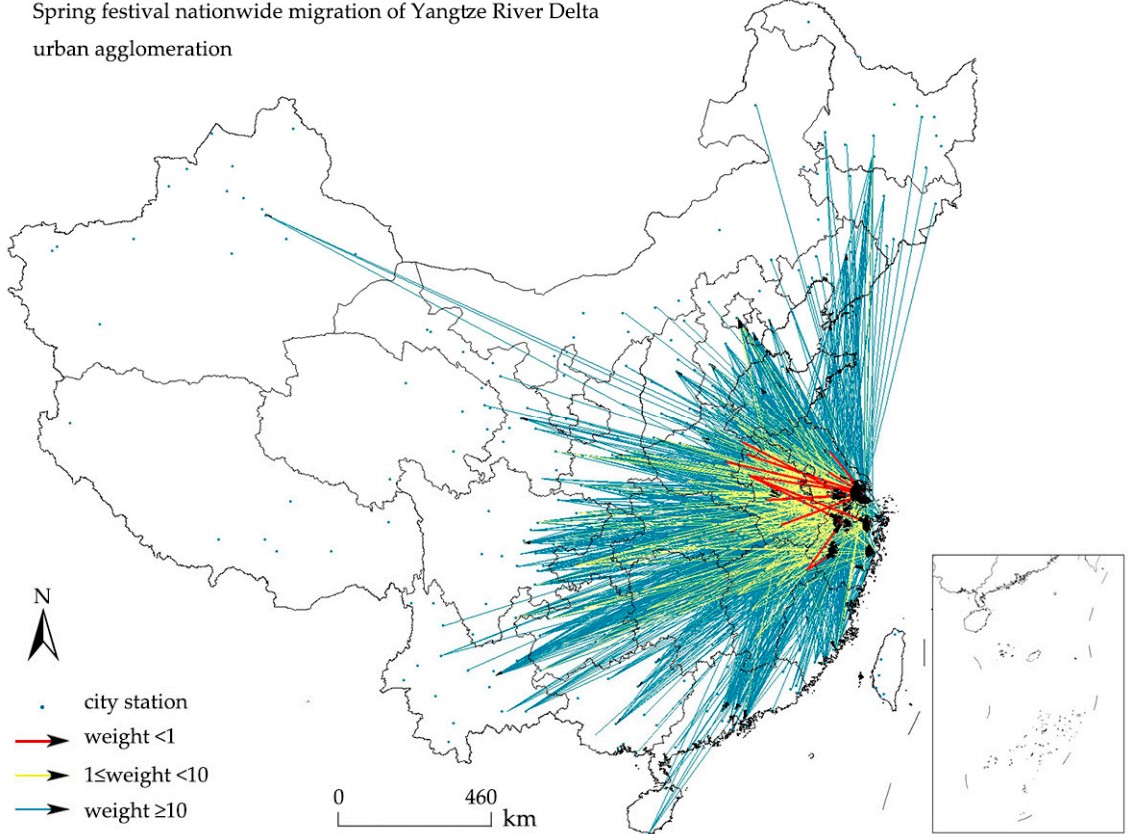

**Figure 2.** Spatial characteristics of population migration of more than 100 people during the Spring Festival travel rush.

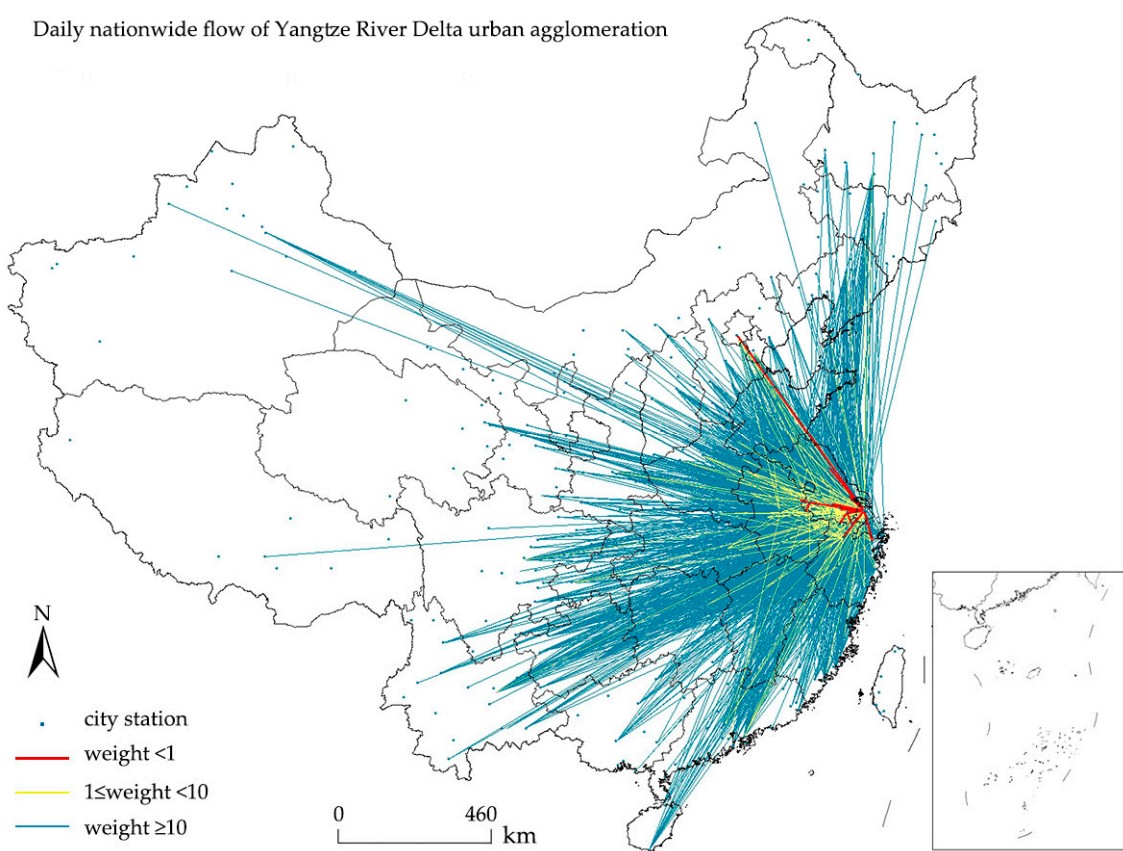

**Figure 3.** Spatial characteristics of population migration of more than 100 people during the daily travel rush.

### 3.2. Internal Relationships of Population Resettlement

During the Spring Festival travel rush, the net migration intensity of the population within the Yangtze River Delta urban agglomeration was 2.86 million, forming a total of 230 city-related items. Among them, Shanghai attracted 1.49 million net migrants, accounting for 52.04% of the total. Suzhou and Nanjing gathered 10.61% and 9.45% of the total population, respectively. Hangzhou, Ningbo, Wuxi, and Changzhou attracted a total of 425,100 people from 59 cities. So far, 86.95% of the incoming population has been accumulated in the seven aforementioned cities. Among the top ten population migration flows with the intensity of net migration, seven link people moving into Shanghai, while five link people moving out of Jiangsu and Anhui, respectively (Table 2). Thus, it can be seen that during the Spring Festival travel rush, the net migration intensities of Jiangsu to Shanghai, Anhui to Jiangsu, and Anhui to Shanghai are relatively strong, and the transboundary migration intensity of Zhejiang is relatively small. During the daily period, the population flow intensity within the Yangtze River Delta urban agglomeration was 5.50 million people, and a total of 231 intercity population mobility correlations were formed. There are nine links of population mobility between Shanghai and the cities of Jiangsu province, representing 2.25 million instances of population mobility, accounting for 40.83% of the total mobility in the same period. The scales of population flow across the three inter-provincial boundaries of Shanghai–Zhejiang, Jiangsu–Anhui and Jiangsu–Zhejiang were 0.99, 0.91, and 0.90 million, accounting for 18.09%, 16.70%, and 16.33% of the total, respectively. The population mobility between Zhejiang and Anhui provinces was 229,500 people, accounting for 4.17% of the total scale. Because Anhui province and Shanghai are separated by the Jiangsu and Zhejiang provinces, they are far apart in space, and the population flow is the smallest, with only 213,100 people. Most of the cities in the top ten daily flow intensities are adjacent and belong to provinces (cities) on both

sides of the inter-provincial boundary, indicating that the population flow intensity is affected by the inter-provincial boundary to some extent.

**Table 2.** Top ten net move-in intensities of Spring Festival and daily flow intensity transport (ten thousand people).

| Spring Festival Net Move-In Travel Rush | Intensity | Daily Flow | Intensity |
|---|---|---|---|
| Nantong → Shanghai | 20.33 | Shanghai–Suzhou | 118.96 |
| Yancheng → Shanghai | 18.65 | Nantong–Shanghai | 29.12 |
| Chuzhou → Nanjing | 13.17 | Jiaxing–Shanghai | 28.23 |
| Anqing → Shanghai | 11.03 | Hangzhou–Shanghai | 26.41 |
| Suzhou → Shanghai | 10.69 | Chuzhou–Nanjing | 23.10 |
| Taizhou-JS → Shanghai | 10.49 | Jiaxing–Suzhou | 20.17 |
| Yangzhou → Shanghai | 8.09 | Shanghai–Wuxi | 19.65 |
| Chuzhou → Shanghai | 7.61 | Ma'anshan–Nanjing | 15.23 |
| Anqing → Suzhou | 7.13 | Nanjing–Shanghai | 14.52 |
| Chuzhou → Suzhou | 6.88 | Ningbo–Shanghai | 13.54 |

By comparing and analyzing the population flow and migration network within the Yangtze River Delta urban agglomeration in two periods (Figure 4), obvious hierarchical characteristics are presented. During the Spring Festival travel rush, there are six first-level population net migration relationships, carrying 0.84 million people, where the migration destinations are Shanghai and Nanjing. There are 56 secondary net migration relationships, carrying 1.64 million people. The destinations include 13 cities, such as Shanghai, Suzhou, and Hangzhou. The 141 third-level population net migration relationships carry 0.38 million people. The destinations include Ningbo, Jiaxing, Wuxi, and another 21 cities. It can be seen that in the net migration network of the Yangtze River Delta urban agglomeration, Shanghai has become the first-level population migration destination, and the second is the Jiangsu province. During daily life, 13 first-level population flows carry 3.43 million people. There are 51 secondary population flows that carry 1.68 million people. The other 154 third-level population flows carry 0.38 million people. During the Spring Festival travel rush, the net migration scale of the three levels of the network accounts for 29.47%, 57.31% and 13.22% of the total migration scale, respectively, and the migration scale of the three levels of the network accounts for 62.42%, 30.62% and 6.95% of the total flow scale, respectively, in the daily period. It can be seen that the secondary network relationship during the Spring Festival travel rush is the main migration path, while the first-level network relationship in the daily stage is the main flow path. The first-level network relationship mainly links Shanghai. As the core city of the Yangtze River Delta urban agglomeration, Shanghai is the most attractive city with the highest level of development and of course the highest cost of living. Almost all of the population migrants during the Spring Festival travel mentioned above are employees from other places. Most of these people represent rural population flows to cities for employment. When choosing a city for employment, these people consider not only employment opportunities and labor benefits, but also the cost of living and the resulting final benefits. Affected by this factor, most migration flows tend to avoid cities such as Shanghai and this has led to secondary network relationships becoming the main migration path.

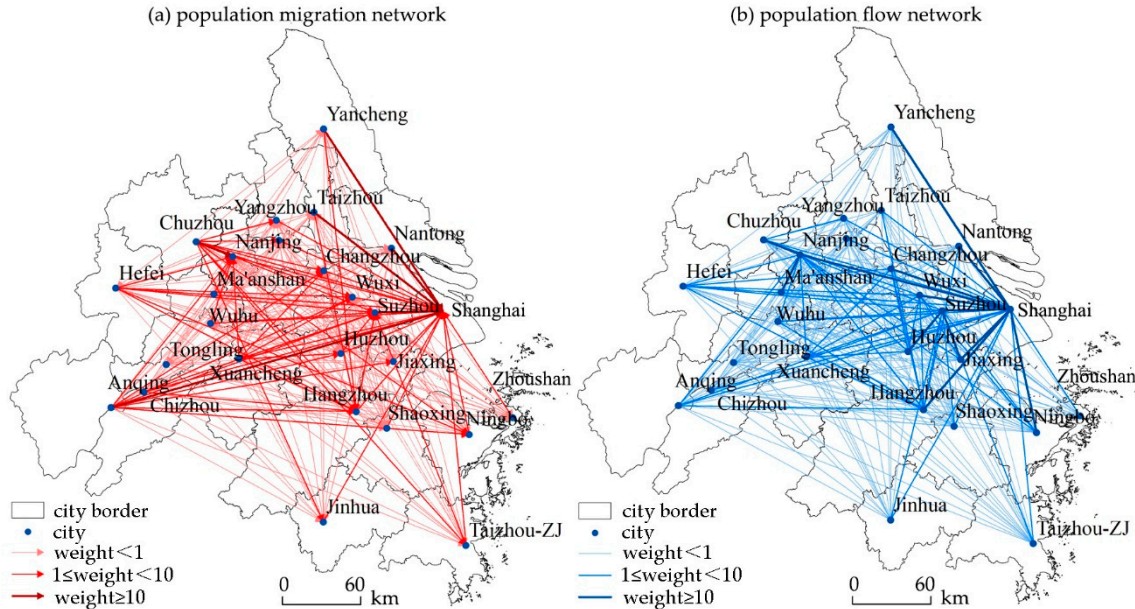

**Figure 4.** Spatial characteristics of Spring Festival migration (**b**) and daily flow (**a**) in the Yangtze River Delta urban agglomeration.

### 3.3. Structural Characteristics of Population Flow and Migration Network

During the Spring Festival travel rush, the mean centrality of the net population immigration within the Yangtze River Delta urban agglomeration was 23.08, which means that the 10 cities with net population immigration absorbed an average of 0.23 million people from the Yangtze River Delta urban agglomeration. Shanghai, Nanjing, Suzhou, and Hangzhou rank as the top four for centrality (Table 3). The mean centrality of the net emigration of the population during the Spring Festival travel rush is −14.43, which means the net emigration of the population from the 16 cities in the Yangtze River Delta urban agglomeration is 0.14 million on average. The top four cities are Chuzhou, Anqing, Xuancheng, and Yancheng in terms of the order of centrality. In the daily period, the Yangtze River Delta urban agglomeration's average control force of the network is 42.31, the association hosting an average floating population of 0.42 million for each population flow link, with the top five cities (Shanghai, Suzhou, Nanjing, Hangzhou, and Jiaxing) carrying 65.8% of the total population flow, of which Shanghai represents 31.4% of the daily population flow intensity of the Yangtze River Delta urban agglomeration. By calculation, the network density of the Yangtze River Delta urban agglomeration based on the daily population flow is 0.711, reaching the percolation threshold of 0.5, and the overall network connection shows a strong state of connection. In contrast, the network density is slightly higher than 0.432 in the middle reaches of the Yangtze River [50]. As one of the most developed regions in China, the Yangtze River Delta urban agglomeration has a high degree of inter-city contact, more ways to obtain information and share resources, and a high degree of reciprocity of network resources, basically forming an integrated urban network.

**Table 3.** Attribute table of city nodes of the population flow and migration network.

| City | Related Cities Number | Centricity | Control | City | Related Cities Number | Centricity | Control |
|---|---|---|---|---|---|---|---|
| Shanghai | 25 | 148.99 | 345.36 | Taizhou-ZJ | 18 | −4.20 | 7.68 |
| Nanjing | 17 | 22.68 | 83.93 | Shaoxing | 18 | −4.40 | 12.96 |
| Suzhou | 17 | 19.69 | 178.46 | Ma'anshan | 18 | −5.36 | 18.02 |
| Hangzhou | 18 | 13.44 | 59.75 | Yangzhou | 17 | −7.35 | 15.74 |
| Ningbo | 18 | 8.46 | 23.24 | Wuhu | 18 | −10.24 | 12.72 |
| Wuxi | 17 | 7.87 | 37.32 | Chizhou | 18 | −11.45 | 3.60 |
| Changzhou | 17 | 3.66 | 22.47 | Taizhou-JS | 17 | −12.99 | 12.02 |
| Jiaxing | 18 | 3.38 | 56.25 | Hefei | 18 | −16.51 | 25.11 |
| Huzhou | 18 | 2.05 | 34.03 | Nantong | 17 | −21.39 | 34.43 |
| Jinhua | 18 | 0.62 | 10.36 | Yancheng | 17 | −22.50 | 13.58 |
| Zhenjiang | 17 | −0.31 | 8.30 | Xuancheng | 18 | −22.95 | 27.63 |
| Tongling | 18 | −1.93 | 1.32 | Anqing | 18 | −41.65 | 8.77 |
| Zhoushan | 16 | −2.62 | 7.96 | Chuzhou | 18 | −44.97 | 38.97 |

Because the density analysis in the social network analysis only focuses on the binary matrix, and as the strength of connection between cities is different, the density analysis can only reflect the characteristics of the urban network to a certain extent. However, "small world analysis" has the feature that most nodes can be reached from any other point through a few correlations, which makes up for the shortage of density analysis. The average clustering coefficient and average path length can reflect whether there is a "small world" effect in the network—that is, how nodes are embedded in their neighbors. The former gives the overall indication of node clustering or agglomeration, and the latter is the average graph distance between all node pairs, where the graph distance of interconnected nodes is 1. If the average clustering coefficient of a graph is significantly higher than the random graph generated by the same number of nodes, and the average shortest path is similar, then the graph is considered to have "small world" characteristics. The average clustering coefficients of the population flow and migration network in the Yangtze River Delta urban agglomeration are 0.290 and 0.583, respectively, both of which are significantly higher than the random network coefficients of 0.029 and 0.058 under the same nodes number. The average path lengths are 1.799 and 1.289, which is close to 1.818 and 1.306. This indicates that there is an obvious "small world" phenomenon in the urban network of the Yangtze River Delta urban agglomeration based on population mobility, and the local network has the characteristics of a cluster structure. Using a modular calculation method (a community detection algorithm [51]) to measure the cities' association in the Spring Festival travel rush and daily period of the Yangtze River Delta urban agglomeration, four and three cluster structures were obtained, respectively (Figure 5). The population migration network of the Spring Festival travel rush includes the central and southern unidirectional cluster of the Yangtze River Delta, the Jiangsu and Anhui inter-provincial boundary cluster, the core–edge cluster in the Yangtze River Delta, and the Zhejiang and Anhui inter-provincial boundary Huzhou–Xuancheng cluster. The daily population flow network forms the Jiangsu and Anhui inter-provincial boundary cluster, boundary cluster of the Jiangsu, Zhejiang, and Anhui provinces, and the core–edge cluster of Yangtze River Delta. These cluster cities have a closer relationship and are geographically closer to each other, which indicates that the urban closeness based on population migration still follows the restriction of the geospatial effect. The core–edge clusters in the Yangtze River Delta are spatially divided by other cluster structures, showing a spatial jump, and are distributed on the east and west sides of the Yangtze River Delta. It can be seen that under the background of flow space, population flow and migration have a tendency to partially overcome spatial friction.

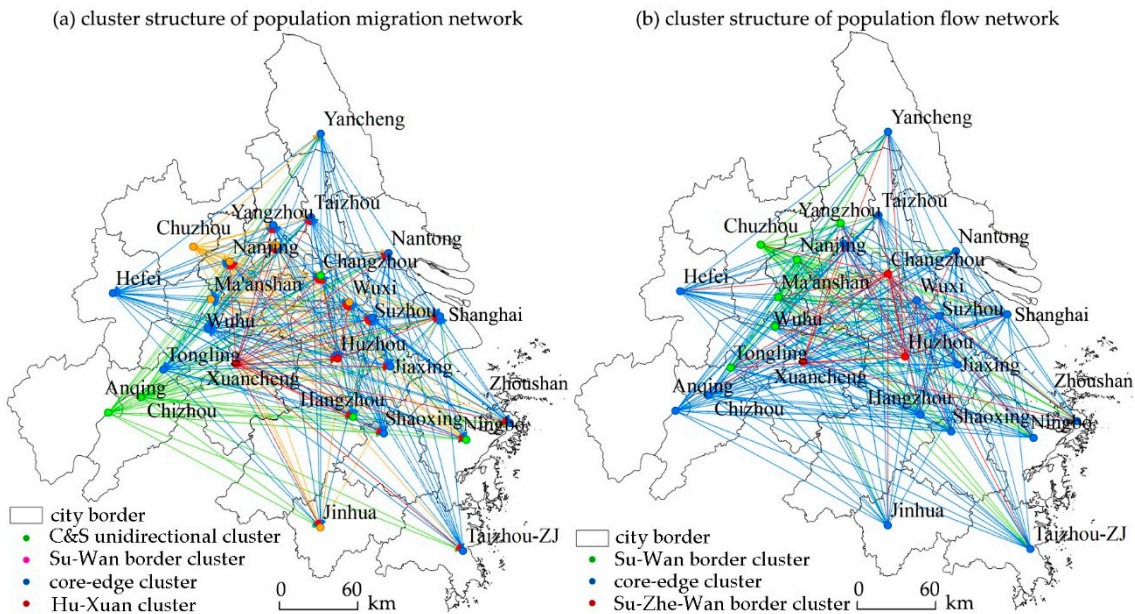

**Figure 5.** Population flow (**b**) and migration (**a**) network cluster structure diagram of the Yangtze River Delta urban agglomeration.

## 4. Analysis of Influencing Factors of Network Pattern Formation

### 4.1. Variable Selection

In the urban network based on population flow and migration, the relationship of the urban network originates from population flow and migration between provinces, so the path selection behavior of population flow and migration constitutes the microfoundation for understanding the urban network correlation pattern. Based on the analysis framework of "behavior-structure-effect", this paper explains the formation mechanism of urban cyberspace pattern in the Yangtze River Delta urban agglomeration. In the analysis framework, the spatial migration behavior of population at the micro level is manifested as the urban connection process with the main characteristics of path selection and path dependence at the macro level. These dynamic processes promote the orderly evolution of an urban network self-organization system and form certain effects. Therefore, the factors influencing the choice of population flow and migration path fundamentally shape the relationship pattern of the urban network. Based on this idea, the following explanatory variables are selected in this paper:

First, the city attribute index. An urban network based on population flow and migration is more dependent on people's subjective mobility consciousness, which involves organizations' tendency to establish ties with other organizations that can make profits for themselves. The assumption is that being an attractive network partner depends on some individual characteristics of the city. In this paper, five indicators were selected as the proxy variables for urban attributes: (1) economic scale (GDP), which is measured by the GDP per capita of permanent residents in the city region (ten thousand yuan). (2) Political resources (capital), which is represented by the dummy variable of the urban administrative level. The value of municipalities directly under the central government, provincial capital cities, and cities with separate plans is 1, while the values of other cities are 0. (3) Employment opportunities (employment) and the unemployment rate (unemployment) are measured by the newly created urban jobs (%) and the registered urban unemployment rate of permanent residents (%), respectively. (4) Labor wage (wage), using the average wage of employees (yuan) for measurement, and (5) the industrial structure (industry), measured by the proportion (%) of the output value of the secondary and tertiary industries in the gross product.

Second, the geospatial index. The geographical location and traffic conditions are the important factors affecting the intensity of population mobility between cities. In this paper, five indicators were

selected as the proxy variables of geographical space: (1) spatial distance (minimum distance), which is measured by the shortest travel distance (km) between each pair of cities, representing the physical distance in the place space (based on Baidu map query, the shortest traffic distance between each pair of cities in the hybrid transportation system of high-speed railway, ordinary railway, expressway, national highway and provincial highway); (2) time distance (minimum time), measured by the shortest travel time (min) between each pair of cities. This is used to represent the network proximity in the flowing space (through Baidu map query, the shortest travel time between each pair of cities (if transfer is needed, the transfer time is included) was obtained based on the above transportation system. In order to ensure the accuracy of the data, the starting and ending points were selected as the main railway stations of each city in the query process. In addition, 12306 and Gaode auto navigation were used to conduct navigation queries on 233 pairs of urban travel to further verify the reliability of the data); (3) shoreline virtual variable (shoreline), where the code of cities along the Yangtze River and coastal cities shall be 1, and the codes of other cities shall be 0; (4) the border area virtual variable (border), where areas located on both sides of the inter-provincial boundary cities code are assigned 1, while other cities are assigned 0 (riverside and coastal cities include the following: Shanghai, Nantong, Suzhou, Yangzhou, Zhenjiang, Nanjing, Wuhu, Tongling, Anqing, Yancheng, Jiaxing, Ningbo, Zhoushan, and Taizhou-ZJ. Cities along the inter-provincial boundary include the following: Shanghai, Suzhou, Jiaxing, Wuxi, Huzhou, Hangzhou, Yangzhou, Nanjing, Changzhou, Chuzhou, Ma'anshan, and Xuancheng); (5) trans-border frequency (trans-border). Here, we use the number of inter-provincial boundaries for each pair of related city populations that are required to go through to travel in the shortest travel time and count 233 associations. The number of border crossings is at least one and at most three. The last three variables are used to capture the influence of explanatory variables that do not change with time and provide a basis for subsequent effect studies.

Third, the network time lag (lag 2010). In the development of an urban network, path dependence (also called preferential attachment [52,53]) is an important driving mechanism. This paper assumes that the growth and development of urban network link relationships are not only affected by the relevant variables of the same period, but also depend on the link relationship pattern formed in history—that is, the habitus of choice of the migration path. In order to capture the influence of early network development foundations on the current network spatial structure, the urban network relationship matrix in 2010 is included in the explanatory variables (since there are no systematic migration data from Baidu in 2010, the floating population data used in this paper are from the data of the sixth census).

*4.2. Analysis Results*

In the QAP correlation and regression of the directed weighted network in the Spring Festival travel period and the undirected weighted network in the daily period (Table 4), the terms of the economic scale, political resources, employment opportunities, labor wages, industrial structure, virtual variables of the coastal and riverside coastline, virtual variables of the boundary area, and network time lag were all positive and passed the significance test. For example, in the regression analysis of the net migration of the population during the Spring Festival travel rush, the fitting coefficient of employment opportunities was 0.347. In the random rearrangement experiment, the probability P (large) that the fitting coefficient was larger than the actual observed value was 0.025, while the probability P (small) that it was smaller than the actual observed value was 0.975. In the daily population mobility network regression analysis, the fitting coefficient of employment opportunities was 0.405. In the random rearrangement experiment, the probability that the fitting coefficient was greater than the actual observed value was 0.000, while the probability that it was less than the actual observed value was 1.000. The unemployment rate, spatial distance, time distance, and trans-border frequency were all negative values in the regression analysis of the population migration network during the Spring Festival travel rush and the daily population flow network, where neither the unemployment rate nor the spatial distance passed the significance test in the regression analysis of the

population migration network during the Spring Festival. Spatial distance failed the significance test in the regression analysis of the daily population flow network. In a comprehensive sense, 12 factors, including economic scale and spatial distance, explain the poor 85.6% distribution of network link relationships of net migration in the Spring Festival travel rush and the 89.0% value of the floating population for the daily period.

**Table 4.** Correlation and regression analysis results of quadratic assignment procedure (QAP) matrix.

| | Spring Festival Travel Rush | | | | Daily Period | | | |
|---|---|---|---|---|---|---|---|---|
| | R | Fitting Coefficient | P (Large) | P (Small) | R | Fitting Coefficient | P (Large) | P (Small) |
| GDP | 0.334 *** | 0.057 *** | 0.025 | 0.975 | 0.368 *** | 0.071 *** | 0.001 | 0.999 |
| Capital | 0.521 *** | 0.067 *** | 0.100 | 0.900 | 0.454 *** | 0.247 ** | 0.071 | 0.929 |
| Employment | 0.925 *** | 0.347 *** | 0.025 | 0.975 | 0.877 *** | 0.405 *** | 0.000 | 1.000 |
| Unemployment | 0.209 * | 0.262 | 0.722 | 0.278 | 0.255 * | 0.068 * | 0.075 | 0.925 |
| Wage | 0.763 *** | 0.051 *** | 0.000 | 1.000 | 0.735 *** | 0.312 *** | 0.000 | 1.000 |
| Industry | 0.393 ** | 0.030 ** | 0.065 | 0.935 | 0.375 * | 0.075 * | 0.025 | 0.975 |
| Minimum distance | 0.303 * | 0.033 | 0.724 | 0.276 | 0.333 * | 0.064 | 0.866 | 0.134 |
| Minimum time | 0.542 ** | 0.209 ** | 0.066 | 0.934 | 0.653 *** | 0.291 ** | 0.005 | 0.995 |
| Shoreline | 0.210 ** | 0.047 ** | 0.000 | 1.000 | 0.226 ** | 0.051 ** | 0.000 | 1.000 |
| Border | 0.646 *** | 0.174 *** | 0.005 | 0.995 | 0.652 *** | 0.335 *** | 0.000 | 1.000 |
| Transborder | 0.196 *** | 0.031 *** | 0.044 | 0.946 | 0.106 *** | 0.072 *** | 0.022 | 0.978 |
| Lag2010 | 0.903 *** | 0.799 *** | 0.000 | 1.000 | 0.839 *** | 0.701 ** | 0.000 | 1.000 |
| $R^2$ | 0.856 | | | | 0.890 | | | |

Note: ***, ** and * represent significance at the levels of 1%, 5%, and 10%, respectively.

First, the economic scale, political resources, and industrial structure are important factors influencing the formation of a population flow and migration network, while employment opportunities and labor wages are key guiding factors for population migration. Through the hypothesis test of the urban attribute and network relationship, it can be concluded that the expected population flows and migration between high–high and low–high per capita GDP cities are significantly lower than the actual values, and the expected population flow and migration between low–low per capita GDP cities is significantly higher than the actual observed value (the judgment of cities with high and low attribute values was based on the following: the average value of the attribute value was taken as the truncation value. Cities whose actual attribute value was higher than the truncation value were determined as the "high" type. Cities whose actual attribute value was lower than the truncation value were determined as the "low" type). The hypothesis test of the relationship between political resources, industrial structure, and the network presents similar characteristics. The correlation analysis results of the QAP for employment opportunities, labor wages, and network relationships show correlation coefficients between 0.735 and 0.925 and all passed the test of significance here, with highest relevance in the city property index of the six variables, which can provide a key population flow moving direction selection guide. The expected value of population migration between cities with high–high and low–high employment opportunities is significantly lower than the actual observed value, and the expected value of population migration between cities with low–low employment opportunities is significantly higher than the actual observed value; the characteristics of the hypothesis testing results of labor wage and network relationship are the same as above. Although the unemployment rate has a negative impact on the net population migration, it does not constitute the main factor affecting the population flow and migration because it does not pass the significance test of regression analysis. The results show that cities with a large economic scale, rich political resources, and high degree of industrial structure optimization are conducive to promoting population flow and migration. In addition, cities with more employment opportunities and high labor wages are conducive to attracting flow and migration.

Secondly, the spatial distance and time distance are conditional factors that affect the formation of population flow and migration networks. Among them, the QAP correlation analysis results of the time

distance and network relationship all pass the significance testing, while only the Pearson correlation coefficient and gamma value of the correlation analysis results for the spatial distance and network relationship pass the significance testing, indicating that time distance has a more significant impact on the network spatial pattern of population migration. In the hypothesis test between time distance and network relationship, the expected value of population flow between cities with a short time distance was significantly lower than the actual observed value, while the expected value of population flow and migration between cities with long time distance was significantly higher than the actual observed value. Shoreline cities along the river (sea) and border area cities are beneficial to forming closely linked relationships with the population flow. The hypothesis test between these cities and the network relationship shows that shoreline cities along the river (sea) and border area cities have population flow expectations that are significantly lower than the observed values. However, other population flow values for cities that are near the expected value are significantly higher than the observed values. The QAP correlation analysis shows that in the geographic spatial index, the boundary region has the most significant influence on the formation of the network spatial pattern of population migration. According to the statistics here, in the 233 pairs of population flow and migration, the trans-border frequency is restricted by the spatial distance and time distance and has a certain negative influence on population resettlement. The results show that the inter-provincial boundary plays a dominant role in the development of the inter-provincial population flow and migration network relationship. Under the multiple influences of the spatial distance, time distance, and trans-border frequency, whether the optimal path can break through the inter-provincial boundary becomes the decisive factor for the formation of the spatial pattern of the population flow and migration network.

Thirdly, the "inertia flow and migration" caused by the historical basis has a profound influence on the formation and development of the population flow and migration network. The QAP correlation analysis results of the network relationship in 2010 and 2015 show that the correlation coefficient and other participating fitting values are positive and pass the significance test, and the correlation between them is robust. Cities with more linked relationships in history tend to further enhance the strength of said linked relationships, while those without an active population flow and migration history still have fewer link relationships. This indicates that the mechanism of cyclic accumulation is a causal relationship and is an important basis for linked relationships in population migration networks, which leads to the characteristics of path dependence and self-reinforcement in the development of the network patterns. However, the scale and strength sequences of population mobility links in 2010 and 2015 do not correspond completely here, indicating the dynamic nature of urban competitiveness and the inter-city population flow relationship, as well as the complexity of the evolutionary dynamic mechanism of urban network relationships.

## 5. Conclusions and Discussion

### 5.1. Conclusions

Migration data based on the measured flow and migration data of Baidu for the Yangtze River Delta urban agglomeration during the Spring Festival and daily period were used here, analyzing the characteristics and the network structure. Three factors were introduced, including inter-provincial borders, the trans-border frequency, and time distance, which are embedded in traditional network pattern influencing factors. We have discussed the formation of the Yangtze River Delta urban agglomeration's population flow and migration network structure mechanisms; relevant conclusions include the following:

(1) During the Spring Festival travel rush, Shanghai, as a central city, has a significant siphon effect, with Suzhou, Nanjing, Hangzhou, Ningbo, Wuxi, and Changzhou gathering 86.95% of the incoming population. The net migration intensity of Jiangsu to Shanghai, Anhui to Jiangsu, and Anhui to Shanghai was stronger, and the transboundary migration intensity of Zhejiang Province was relatively small. In the daily period, the cross-border population mobility between Shanghai and other

provinces accounted for 40.83% of the total scale of the same period. Most of the top ten associated cities in terms of flow intensity are located adjacent to each other, and they belong to provinces (cities) on both sides of the inter-provincial boundary, indicating that the population mobility intensity is affected by the inter-provincial boundary to a certain extent. The population migration network has obvious hierarchical characteristics. During the Spring Festival travel rush, the secondary network relationship is the main migration path, while the first-level network relationship in the daily stage is the main migration path. This shows that the daily population flow has a significant core spatial agglomeration effect, but the migrant population, who are employed in different places, takes into account the decline in the total employment income caused by the cost of living, and often avoids the first-level network relationship and chooses to move into the secondary core of the urban agglomeration to seek employment opportunities through the secondary network relationship. (2) The values of the network density, mean centrality, and the average control force indices were 0.711, 23.08, and 42.31, and they all show that the Yangtze River Delta urban agglomeration network, based on population flow and migration, presents a strong state of connection. In addition, the calculation results of the average clustering coefficient and average path length of the network nodes show that there is an obvious "small world" phenomenon in the population flow and migration network of the Yangtze River Delta urban agglomeration, and that cluster structure characteristics are formed locally in the network. Using a modular calculation method to measure the Spring Festival travel and daily association, four and three cluster structures were obtained, respectively. Among them, the central and southern parts feature a unidirectional cluster, with a Jiangsu and Anhui inter-provincial boundary cluster, Zhejiang and Anhui inter-provincial boundary Huzhou–Xuancheng cluster, and Jiangsu, Zhejiang, and Anhui inter-provincial boundary cluster. These cities have a closer relationship and are geographically closer to each other, which indicates that the urban closeness based on population migration still follows the geospatial restriction effect. Core-edge clusters are spatially divided by other cluster structures, showing a spatial jump, and are distributed on the east and west sides of the Yangtze River Delta. It can be seen that against the background of flow space, population migration has a tendency to partially overcome spatial friction. (3) Among the traditional factors, economic scale, political resources, and industrial structure are the important influencing factors for the formation of population flows and migration networks, while employment opportunities and labor wages are the key guiding factors for population flow and migration. Spatial distance is a conditional factor that affects the formation of population flows and migration networks. The "inertia flow and migration" caused by a historical basis has a profound influence on the formation and development of population flows and migration networks. The innovatively introduced boundary region variable has a significant positive influence on the pattern of population flow and migration in cyberspace. The degree of influence even exceeds traditional factors such as the economic scale and political resources, and the relevance is second only to employment opportunities and labor income. Due to the restriction of the spatial distance and time distance, the transboundary frequency has a certain negative influence on the network pattern of population flow and migration. The influence of the time distance on the spatial pattern of population flow and migration network is greater than that of the spatial distance, which is a conditional factor that affects the formation of population flows and migration networks. Therefore, the inter-provincial boundary plays a leading role in the development of the inter-provincial flow and migration network relationship. Under the multiple influences of the spatial distance, time distance and trans-border frequency, whether the optimal path can break through the inter-provincial boundary becomes the decisive factor for the formation of a spatial pattern for population flows and migration networks.

## 5.2. Discuss

On the basis of traditional urban network structure influence factors, this paper has introduced three factors, including the time distance, inter-provincial administrative boundary (because of China's special administrative divisions), and trans-border frequency, to quantitatively analyze the Yangtze River Delta urban agglomeration in terms of the population flow and migration network pattern

formation, as well as the impact of the development process. We have also explained the mechanism of action. This not only enriches the factors that influence the formation and development of a network structure, but also verifies the rationality and necessity of adding new factors through matrix correlation and regression analyses to prove the objectivity of the relationship between the urban network and inter-provincial boundary, which means that there is no "false relationship" in terms of statistical significance. The disadvantage is that this paper confirms that the inter-provincial boundary plays a leading role in the formation and development of the inter-provincial population flow and migration network relationship, but it has not yet discussed the boundary effect in the urban network structure in terms of the flow and migration behavior, which will be one of the focuses of follow-up research. In addition, this article has analyzed the population migration network patterns and causes of the Yangtze River Delta urban agglomeration in a specific year, while multi-space, multi-scale, and multi-temporal comparative studies have not yet been performed. Currently, Xuewei Wang is leading a team to conduct related research, including a comparative study of the population migration network patterns and causes of the three major urban agglomerations of the Yangtze River Delta, central China, and Chengdu–Chongqing in the Yangtze River Economic Belt, as well as an analysis of the 2020 Spring Festival and daily population migration and flow patterns. For now, the preliminary results show that there are certain differences in the patterns of population migration networks in different regions and that the impact of factor groups is also different. In addition, due to the impact of coronavirus, introducing an era of pandemic crises, network patterns and influencing factor analysis based on the data from February to April in 2020 show significant abnormalities, indicating that, during a pandemic, the formation and development factors of the population migration network need to be further improved.

**Author Contributions:** Data curation, X.W.; formal analysis, X.W. and D.F.; funding acquisition, W.C.; methodology, X.W. and B.T.; project administration, W.C. and D.F.; writing—original draft, X.W.; writing—review & editing, X.W. and S.D. All authors have read and agreed to the published version of the manuscript.

**Funding:** This research was funded by the National Natural Science Foundation of China, grant numbers 41901151. The APC was funded by the National Natural Science Foundation of China, grant numbers 41901151.

**Conflicts of Interest:** The authors declare no conflict of interest.

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
