# Peer review of "Research on Network Patterns and Influencing Factors of Population Flow and Migration in the Yangtze River Delta Urban Agglomeration, China"

_sustainability, doi:10.3390/su12176803_

Round 1
Reviewer 1 Report
The authors present a very good and extensive research study. The only remark applies to the illustrations - they could be more legible and of better quality.
Author Response
Five illustrations of the article have been reprocessed to ensure better quality and more legible.
The specific revised content can be seen in the revised manuscript(attachment).
Thank you very much for your revision suggestions. Your suggestions have greatly improved the article, and also made me learn a lot. Thanks again and wish you all the best.

Reviewer 2 Report
Writing is not very smooth and contains many typos to review.
The introduction is chaotic and sometimes repetitive. Considering that the topic addressed is complex, it does not predispose to understanding research.
The conclusions are too concise and do not present any output.
Author Response
Response 1: I have reviewed the whole article, made corresponding changes and improvements in logic, language expression, and word spelling. Specific modifications were revised and marked in the revised manuscript.
Response 2: Major revisions and adjustments were made to the introduction part of the article, and redundant content has been deleted. For instance, the expression of “network and administrative boundaries are the two sides faced in the development of regional integration” in the manuscript appeared three times repeatedly, thus was simplified in the revised manuscript, only making the necessary reservation once in the first paragraph. In the revised manuscript, the introduction part consists of four paragraphs. The first paragraph introduces research background, the relationship between the urban network and boundary, theoretical basis and research summary; The second paragraph introduces the selection and analysis of research perspective, namely “population flow and migration” (including daily population mobility and population migration during the Spring Festival); The third paragraph introduces the innovative and scientific aspects of data selection in this article; The last paragraph introduces the research framework and main contributions of this article.
Response 3: Conclusions and discussions of the article were reorganized, the corresponding research results and the scientific issues or realistic conclusions indicated behind the results were supplemented in the three conclusions of the article (The specific content can be seen in the revised draft). Moreover, deficiencies of the research and focus of the next research work were added to the discussion part of the article.
Please see the attachment.
Thank you very much for your revision suggestions. Your suggestions have greatly improved the article, and also made me learn a lot. Thanks again and wish you all the best.

Reviewer 3 Report
To make your article robust and highlight its strength you should provide a better introduction, especially the contribution of this paper. In other words, what a scholar that studies India, Mexico, Brazil, or Congo - countries in the process of industrialization, which shares similar traits of rural-urban migration - can learn from your paper? Why should they read it? Why only a few studies have focused on network relations and factor mobility? What have they missed? Why is it important to include that? To convince your reader, you should articulate an outline of the general debate and point out the contribution of your intervention. Rather than listing, you need to explain. For example, on page 2, you say how different branches of geography, namely urban, economic, and political geography approach their field of study. However, it remains unclear whether you are criticizing or supporting them. If you mention them, you need to explain what you take away from them, how you use them, what are their shortcomings, and how you think this article aims at correcting their weaknesses or using their strengths.
Additionally, it would be great if you include studies on the same topic in other countries and continents. Such a step would strengthen the contribution of your work. By connecting and comparing the Yangtze case-study with other similar cases would deprovincialize this work and include it into a larger debate. The dynamics that this article explores are not unique to China and for this reason, it is important to compare and integrate your data with that of other works that investigate similar processes.
You mention the way globalization is intertwined with interregional integration. However, you stop short here. I think this is very important. Although this is not the focus of your article, you need to explain better the correlation between internal migration networks and transnational networks.
Spring festival represents a point of interest for investigating the flows of human movement across time and space and reconstructing the transregional migration. The question that arises is: and so what? What we learn from your conclusion that the secondary network relationships are the main migration paths. Why this conclusion is important and what do we learn for it? What does it tell as regarding the importance of the regional boundaries as important variables that define migrant networks and flows? Why do you choose the Spring festival? There might be other important events or variables to explore these networks and how the regional boundaries affect or are affected by them. What is the rationale of your choice? How does this relate to your methodology?
Although at first glance it might seem that considering the ramifications of the trans-regional migration networks falls outside of your academic scope, I'd suggest considering them - at least briefly. This becomes true in an era of pandemic crises. How migration paths carry pathogens, capital, ideas across boundaries. After all, the boundaries become real or fictitious when they are juxtaposed to the ways people respect or disrespect them. They can serve as an example that supports your argument - or destabilizes it - on the leading role regional boundaries play in the migrant flows.
Author Response
Response 1: First of all, I reorganized the writing of the introduction content and adjusted the logic, In revised version, reference significance and main contributions of this research are introduced respectively in the second paragraph of research perspective selection and the fourth paragraph of research framework combing (The specific content can be seen in revised draft); Secondly, in the first section of the introduction, especially the research progress of the boundary, the reason why boundary research pays less attention to network relationship and element mobility was explained. The specific supplementary content is “because boundary discussion originated in the field of economic, and the relative flow data between administrative regions is difficult to obtain, there are few researches based on the perspective of element flow network relations, more attention is paid to the problem of differentiated development on both sides caused by boundary that as administrative barriers. Nowadays, a large amount of behavioral spatiotemporal labeled data with individual granularity helps to compare whether cross-border activities are inhibited based on the perspective of element flow”; Thirdly, it summarizes the deficiencies and shortcomings of existing research basis of the progress of boundary research, and puts forward the necessity of a boundary and network overlay research system.
Response 2: Currently, I am leading my team on two tasks. One is to do a comparative study on the characteristics and causes of the population flow and migration network between different regions in China, and the quantitative analysis has been completed so far; the other is using similar research perspectives and methods to conduct related research on population mobility network in London urban agglomeration, and the current work is to collect data with the teachers of Cardiff University. Due to the limitation of article length and work progress, these researches have not been added to this article. If the reviewer is interested in this part of research content, please do not hesitate to contact me at any time. I am willing to share the research materials and contents. My email address is wangxw697@ahnu.edu.cn. Additionally, this paper is an important part of the National Natural Science Foundation of China (41901151) hosted by me, so I will actively absorb the opinions of experts, continue to complete a series of research, and finally summarize them into a book.
Response 3: With the continuous advancement of globalization and regional integration, regional network, as an important part of the country and global network, make the development of regional network within the country become the core content of global network construction, which implicates that internal migration network is the important components of transnational network, as well as the basic unit of transnational network’s construction. However, diversified boundary research perspectives often only focus on the phenomenon of regional differentiation caused by the objective existence of boundaries only, with the lack of construction path and development mechanism of the international network embedded in national boundaries under the background of globalization and regional networks embedded in inter-provincial boundaries under the background of regional integration. Therefore, it is particularly important to conduct research on the network structure and its causes in this context.
Response 4: The reasons for choosing Spring Festival are as follows, as a populous country, the research topic of population flow and migration in China is of great significance. driven by multiple factors of economy, environment, policy, and culture, the scale and characteristics of the Spring Festival migration are completely different from the daily population flow. Compared with the population flow of other countries, it also constitutes a research perspective of network unique to China. The comparative study between daily population flow and population migration during the Spring Festival is particularly important; Secondly, the quantitative analysis results of the article show that the first-level network is the main route of daily population flow, while the secondary network is the main route of population migration during the Spring Festival. This conclusion not only highlights the differentiation of the Spring Festival travel and daily network patterns under the background of the Chinese traditional consciousness and cultural, but also reveals that long-distance and long-period population migration often considers the final benefits, so the path selection of the population migration during the Spring Festival will mostly avoid first-tier cities such as Beijing and Shanghai, and these cities are precisely the core nodes of the first-level network relationship; Thirdly, in the selection of factors affecting network formation and development, it is mainly considered that the basis of network construction is the cross-border flow of elements, whose premise is to break through the barriers of administrative boundaries. Therefore, the study of network structure should not lack the discussion of boundary influences (boundaries effect), which provides the basis to the research on the optimization and upgrading of the network structure under the influence of the boundary. In terms of methodology, it is through the matrix correlation and regression analysis of QAP to realize the quantitative comparison of network influencing factors, and also through this method, it is measured that the boundary variables have indeed played a very important role in formation and development of the network.
Response 5: The article analyzed the population migration network pattern and causes of the Yangtze River Delta urban agglomeration in a specific year. However, the disadvantage is that a multi-space, multi-scale, multi-temporal comparative study has not yet been formed. Currently, I am leading a research team to conduct related research, including the comparative study of the population migration network patterns and causes of the three major urban agglomerations of Yangtze River Delta, Central China, and Chengdu-Chongqing in the Yangtze River Economic Belt, as well as the analysis of the 2020 Spring Festival and daily population migration and flow patterns, preliminary results have shown that there are certain differences in the pattern of population migration networks in different regions, and the impact of factor groups is also different. In addition, due to the impact of the coronavirus in an era of pandemic crises, the network pattern and influencing factor analysis results based on the data from February to April 2020 show significant abnormalities, indicating that in era of the pandemic crisis. the formation and development factors of the population migration network need to be further improved. This suggestion is really a great inspiration, providing a very good idea for my follow-up research.
Please see the attachment for details.
Thank you very much for your revision suggestions. Your suggestions have greatly improved the article, and also made me learn a lot. Thanks again and wish you all the best.

Round 2
Reviewer 2 Report
The reading is very heavy because there is an excessive use of analytical and fractional interpretation of the data, some of which are irrelevant.
In particular, the reading is not easy because the consequentiality of the concepts expressed is not understood. It seems that many different phrases have been approached without proper mediation.
Furthermore, considering that a very specific issue is being addressed, it would be useful to investigate if there are other similar cases in the world with which to compare.
Author Response
Response 1: Part of the content of the data analysis in article has been deleted and condensed, especially the sections 3.1 and 3.2, because this article mainly discusses the network structure characteristics and influencing factors of the inter-provincial population flow and migration in the Yangtze River Delta urban agglomeration. The manuscript contains too much content in the analysis of the external network relationship and network description of the urban agglomeration, so the revised draft has been simplified.
Response 2: Due to the inconsistent expression of individual phrases during translation, the professional concepts and phrases of the article are redundant, which creates a reading burden for review experts and readers. During this round of revisions, I have purchased the language editing service recommended by the MDPI system. Professionals sorted out the language of the article to further avoid redundancy and complicated expressions.
Response 3: In this article, city network involves two kinds of population mobility issues, one is the urban-rural migration during the Spring Festival travel period, and second is the commuting movement during the daily period. Regarding the urban-rural migration of population, in India, Mexico, Brazil and other countries, the urban-rural population migration during the industrialization development period has similar characteristics, can be compared for research, which is mentioned in the second paragraph of the article introduction; For the daily commuting flow of population, some countries around the world exist similar mobility phenomena. I am studying the daily population mobility in the Greater London area of the United Kingdom, and expected to compare it with the research results of the Yangtze River Delta urban agglomeration in China. Because it is not yet certain whether the research results of the two urban agglomerations are suitable for comparative research, we have to wait for the conclusion of the UK case study before starting the comparative study.
Thank you very much for the second round of review. Related comparative studies will continue, please stay tuned. Thanks again and wish you all the best.
Reviewer 3 Report
I saw your cover letter. I think it would be great if you include your answers - of course, condensed - in the article. I'd also strongly suggest the authors make language and stylistic editions - mainly punctuations and repetitions of the same words in the same sentence. Some of the sentences are too long - it would be great if you break them down.
Author Response
During this round of revisions, some answers of round 1 have been condensed in the article. Additionally, I have made some language and expression editions, including punctuation changes, deletion of repeated words in the same sentence, etc. Additionally. I have purchased the language editing service recommended by the MDPI system. Professionals sorted out the language of the article to further improve article quality.
Thank you very much for the second round of review, and wish you all the best.
